# Differential personality change earlier and later in the coronavirus pandemic in a longitudinal sample of adults in the United States

**Angelina R. Sutin**[1]*, **Yannick Stephan**[2], **Martina Luchetti**[1], **Damaris Aschwanden**[1], **Ji Hyun Lee**[3], **Amanda A. Sesker**[1], **Antonio Terracciano**[1]

**1** Florida State University College of Medicine, Tallahassee, FL, United States of America, **2** University of Montpellier, Montpellier, France, **3** University of Michigan, Ann Arbor, MI, United States of America

* angelina.sutin@med.fsu.edu

**Data Availability Statement:** The data used in the current analyses can be downloaded from: https://uasdata.usc.edu/index.php?r=

## Abstract

Five-factor model personality traits (neuroticism, extraversion, openness, agreeableness, conscientiousness) are thought to be relatively impervious to environmental demands in adulthood. The coronavirus pandemic is an unprecedented opportunity to examine whether personality changed during a stressful global event. Surprisingly, two previous studies found that neuroticism decreased early in the pandemic, whereas there was less evidence for change in the other four traits during this period. The present research used longitudinal assessments of personality from the Understanding America Study (N = 7,109; 18,623 assessments) to examine personality changes relatively earlier (2020) and later (2021–2022) in the pandemic compared to pre-pandemic levels. Replicating the two previous studies, neuroticism declined very slightly in 2020 compared to pre-pandemic levels; there were no changes in the other four traits. When personality was measured in 2021–2022, however, there was no significant change in neuroticism compared to pre-pandemic levels, but there were significant small declines in extraversion, openness, agreeableness, and conscientiousness. The changes were about one-tenth of a standard deviation, which is equivalent to about one decade of normative personality change. These changes were moderated by age and Hispanic/Latino ethnicity, but not race or education. Strikingly, younger adults showed disrupted maturity in that they increased in neuroticism and declined in agreeableness and conscientiousness. Current evidence suggests the slight decrease in neuroticism early in the pandemic was short-lived and detrimental changes in the other traits emerged over time. If these changes are enduring, this evidence suggests population-wide stressful events can slightly bend the trajectory of personality, especially in younger adults.

## Introduction

Since the beginning of the coronavirus pandemic, there has been interest in tracking its effect on psychological outcomes [1]. This published work has focused understandably on factors

eNpLtDKyqi62MrFSKkhMT1WyLrYyNAeyS5Ny
MpP1UhJLEvUSU1Ly80ASQDWJKZkpIKax
IZKlhYmSdS1cMG0-Euo. Note that data are
available but users must first register for a free
account with UAS before the link will direct them to
the dataset for download. The analytic scripts are in
supplementary material.

**Funding:** Research reported in this publication was
supported by the National Institute on Aging of the
National Institutes of Health under Award Number
R01AG053297 to ARS. The content is solely the
responsibility of the authors and does not
necessarily represent the official views of the
National Institutes of Health. The funders had no
role in study design, data collection and analysis,
decision to publish, or preparation of the
manuscript.

**Competing interests:** The authors have declared
that no competing interests exist.

related to mental health. Many studies, for example, examined how symptoms of depression and anxiety [2], loneliness [3, 4], and social support [5] changed compared to before the pandemic. In addition to aspects of mental and social well-being, the pandemic may have had an impact on more general ways of thinking, feeling, and behaving (i.e., personality). The five-factor model (FFM) [6] of personality operationalizes trait psychological function along five broad dimensions: neuroticism (the tendency to experience negative emotions and vulnerability to stress), extraversion (the tendency to be talkative and outgoing), openness (the tendency to be creative and unconventional), agreeableness (the tendency to be trusting and straightforward), and conscientiousness (the tendency to be organized, disciplined, and responsible). These traits are relatively stable over time [7] but are theoretically thought to be responsive to environmental pressures [8], including stressful events. The coronavirus pandemic has offered the unique opportunity to examine how a global stressful event experienced by the whole population may change personality.

Previous research suggests that personal, but not collective, stressful events may be associated with personality change. Neuroticism, for example, has been found to increase after personal stressful [9, 10] or traumatic [11] events. In contrast, collective stressful events, such as natural disasters, seem to be unrelated to personality change [12, 13]. A study that examined personality change from before to after the 2011 earthquake in Christchurch, New Zealand, for example, found no change in any of the five traits from before to after the disaster (there was a slight increase in neuroticism among participants directly affected by the quake; [12]). In addition, in a sample measured twice after exposure to Hurricane Harvey, there was no evidence of mean-level change in any of the five traits, even among participants with the most exposure [13]. This literature thus suggests that personality traits are not responsive to natural disasters.

In contrast to natural disasters, which tend to be limited in geographic area, the coronavirus pandemic has affected the entire globe and nearly every aspect of life. There is a developing literature on how the pandemic might be shaping personality change. Early in the pandemic, during the acute phase, we examined personality change in a sample of adults from across the United States (ages 18–90). We hypothesized that neuroticism would increase because of pandemic-related stressors and the accompanying fear and uncertainty would lead to more feelings of emotional instability [14]. Surprisingly, however, neuroticism declined slightly between January/February 2020 and March 2020. Although surprising, it is consistent with anecdotal evidence that anxiety (one core aspect of neuroticism) declined early in the pandemic among individuals who typically suffer from anxiety [15]. Further, a small sample from Germany found that neuroticism was slightly lower among university students during the first coronavirus lockdown compared to their neuroticism measured before the pandemic [16]. Although modest, this current evidence suggests that, at least early in the pandemic, during the acute phase, there was a decline in neuroticism.

There is less evidence for change in the other traits from pre- to during the pandemic. Although extraversion was hypothesized to decline because pandemic restrictions (e.g., lockdowns, social distancing, event cancellations) reduced the ability to be sociable, the evidence is mixed: Extraversion decreased slightly in a sample of university students in Germany [16], whereas it did not change in a nationwide sample of adults in the United States accounting for sociodemographic characteristics [14]. No change was found for Openness, Agreeableness, and Conscientiousness in the American sample [14], and these traits were not measured in the German sample [16].

These two studies provided important insights into the early effect of the pandemic on personality. The present research builds on these initial findings in four critical ways. First, we seek to replicate the finding that neuroticism declined early in the pandemic in a larger

national sample of adults in the United States. Second, we address whether the other traits changed in this larger and more diverse sample than the previous samples. Third, with assessments of personality in both 2020 and in 2021–2022, we evaluate differential patterns of personality change across the acute (2020) and adaptation (2021–2022) phases of the pandemic. Finally, with a relatively diverse sample, we test whether personality change was moderated by age, gender, race, Hispanic/Latino ethnicity, or education.

To put any potential change in personality in context, previous research has found that personality changes, on average, about one-tenth of a standard deviation per decade of adulthood [17]. Regarding direction, neuroticism, extraversion, and openness tend to decline from younger to older adulthood, and agreeableness and conscientiousness tend to increase, although neuroticism and conscientiousness may change direction and increase and decrease, respectively, in older adulthood [17]. Although personality traits may change more in younger and older adulthood, compared to middle adulthood, we do not make specific predictions about age differences in personality change during the pandemic because the virus and the response to it has been unprecedented and its effects significant but different across age groups. Older adults, for example, faced a greater threat of severe disease and death, whereas younger adults faced more restriction on age-normative activities. If any differences are found, it would suggest a fruitful future direction to pursue to identify theoretical and empirical reasons for differential personality change by age. If changes are similar across age, it would suggest that personality is reactive to a global stressful event regardless of specific age-related stressors.

The purpose of this research is to examine personality change during the coronavirus pandemic compared to pre-pandemic levels using longitudinal assessments of personality from the Understanding America Study (UAS) [18]. We construe these analyses as exploratory because this study will be the first study of change in personality measured relatively earlier (acute phase) and relatively later (adaptation phase) in the pandemic (pandemic assessments in 2020 and 2021–2022), and because previous findings were not consistent with theoretical expectations. We do expect, however, that neuroticism declined early in the pandemic because of the two previous studies. If this decline is apparent in the UAS sample, it will provide robust evidence that neuroticism was reactive to the pandemic. We do not expect change in the other four traits early in the pandemic based on our previous findings [14]. We do not make predictions about change in personality later in the pandemic or how change may differ by sociodemographic characteristics.

## Materials and methods

### Participants and procedure

UAS is an internet panel study of participants across the United States administered by the University of Southern California [18]. Participants completed surveys through the device of their choice (desktop, laptop, mobile, etc.) and, when necessary, were provided with a device and internet access to participate. To date, the UAS has administered the same personality measure three times (UAS1, UAS121, UAS237). Personality in UAS1 was collected between May 2014-March 2018, personality in UAS121 was collected between January 2018-April 2020, and personality in UAS237 was collected between April 2020-February 2022 (see COVID section below for how assessments were categorized for analysis). All participants had personality measured at least once prior to the pandemic. Because of the sampling structure of UAS, participants reported on their personality again in either 2020 or 2021–2022, but did not report on their personality in both years. As such, for all participants there is one assessment of personality during the pandemic; all available personality data was used in the analyses. Documentation for each wave can be found at the UAS website: https://uasdata.usc.edu/index.php

under "Surveys" and UAS1, UAS121, and UAS237. Participants were included in the analytic sample if they had personality data reported during the pandemic and at least one personality assessment prior to the pandemic. Participants also needed to have sociodemographic information available. A total of 7,109 participants met these criteria, for a total of 18,623 assessments (Mean = 2.62 assessments/participant, SD = .48; range = 2–3; n = 4,495 at UAS1, n = 7,019 at UAS121, n = 7,109 at UAS 237). The current analyses were based on publicly-available, de-identified data and thus did not require approval from the local IRB. The primary data collection was overseen by the IRB at the University of Southern California and written informed consent was obtained from participants. Detailed information about the original data collection, ethical oversight, and consent process can be found in Laith and colleagues [18].

The analyses in this paper were not preregistered and are exploratory. The data used in the current analyses can be downloaded from: https://uasdata.usc.edu/index.php?r=eNpLtDKyqi 62MrFSKkhMT1WyLrYyNAeyS5NyMpP1UhJLEvUSU1Ly80ASQDWJKZkpIKaxlZKlhYmS dS1cMG0-Euo. Note that data are available but users must first register for a free account with UAS before the link will direct them to the dataset for download. The analytic scripts are in supplementary material.

## Measures

**Personality traits.** Personality was measured with the 44-item Big Five Inventory (BFI) [19] at each personality assessment. Participants rated items that measured neuroticism (e.g., can be moody; eight items), extraversion (e.g., is talkative; eight items), openness (e.g., has an active imagination; ten items), agreeableness (e.g., is generally trusting; nine items), and conscientiousness (e.g., is a reliable worker; nine items). Items were rated from 1 (*strongly disagree*) to 5 (*strongly agree*), reverse scored when necessary, and the sum taken in the direction of the domain label (e.g., higher scores on neuroticism indicated greater neuroticism). Although sum scores can sometimes be problematic for missing data, at each personality assessment, more than 99% of participants who completed the assessment had personality scores, which indicated that missing data were not a problem in this study. Scores on neuroticism and extraversion could range from 8 to 40, scores on openness could range from 10 to 50, and scores on agreeableness and conscientiousness could range from 9 to 45. The test-retest correlation between the first and last personality assessment was .72 for neuroticism, .78 for extraversion, .73 for openness, .65 for agreeableness, and .69 for conscientiousness, indicating relatively high rank-order stability, which is similar to test-retest correlations reported during non-pandemic times: .71 for neuroticism, .79 for extraversion, .79 for openness, .70 for agreeableness, and .70 for conscientiousness (Hampson & Goldberg, 2006) [20]. There were some differences between participants who reported on their personality in 2021–22 versus 2020. Specifically, participants who reported on their personality in 2021–2022 were younger at baseline ($d = .28$, $p < .01$), had more years of education ($d = .12$, $p < .01$), were less likely to be men ($\chi^2 = 10.51$, $p < .01$) or Hispanic ethnicity ($\chi^2 = 307.99$, $p < .01$), and more likely to be Asian ($\chi^2 = 111.25$, $p < .01$) than participants who reported on their personality in 2020. After accounting for sociodemographic differences, participants who reported on their personality in 2021–2022 were lower on baseline neuroticism ($d = .08$, $p < .01$) and baseline conscientiousness ($d = .10$, $p < .01$) compared to participants who reported on their personality in 2020.

**Sociodemographic covariates.** Sociodemographic factors were age in years at the first personality assessment, gender (0 = women, 1 = men), race (three dummy-coded variables that compared Black = 1, Asian = 1, and Otherwise-identified = 1 to white = 0), Hispanic/

Latino ethnicity (1 = Hispanic or Latino ethnicity, 0 = not Hispanic or Latino ethnicity) and education, reported on a scale from 1 (*less than first grade*) to 16 (*doctorate degree*). These covariates were selected because of potential age, gender, and education differences in personality and the differential effect of the pandemic across sociodemographic groups.

**COVID.** A variable was created that indicated whether each personality assessment occurred before or during the coronavirus pandemic. We set March 1, 2020 as the start of the pandemic for this sample because widescale closures and cancellations started to occur in the United States in early March 2020. Because patterns of personality change might be different depending on phase of the pandemic, we categorized the COVID personality assessments into two time periods: personality assessments during 2020 (March 1, 2020-December 31, 2020; the acute phase) and personality assessments after 2020 (January 1, 2021-February 16, 2022; the adaptation phase).

### Statistical approach

The trajectory of each personality trait was modeled on time. Time in years was calculated from the first personality assessment to each subsequent assessment. Multilevel modeling (MLM) was used to estimate the trajectory of personality over time (which represents normative developmental/age-related change over time), with random effects for intercept and slope. Level 1 was repeated assessments of personality nested within-person. Socio-demographic variables were entered at level 2 to control for age, gender, race, Hispanic/Latino ethnicity, and education. Following previous studies on pandemic-induced change in subjective age [21] and well-being [22] compared to pre-pandemic levels, we specified a change component that was a time-varying dummy variable that compared all personality assessments prior to the pandemic (May 2014-February 2020; pre-pandemic personality) to the personality assessments during the pandemic (March 2020-February 2022; which reflect normative history-related change during the pandemic). Two dummy-coded variables were created. The first coded personality measured between March 1, 2020 and December 31, 2020 during the acute phase of the pandemic as 1 and others as 0. The second coded personality measured between January 1, 2021 and February 16, 2022 as 1 and others as 0. All participants in the analytic sample had one or two personality assessments prior to the pandemic and one personality assessment in either 2020 or 2021–2022 (because of the structure of UAS, personality assessments were not from the same participants in both 2020 and 2021–2022). The dummy-coded variable indicated whether personality measured during the pandemic increased or decreased during the pandemic compared to pre-pandemic levels.

We also tested whether personality change was moderated by sociodemographic factors (age, gender, race, Hispanic/Latino ethnicity, education) by including an interaction term between each dummy-coded COVID variable and the sociodemographic factor in separate regressions for each interaction. For age, we also ran the same MLM analysis separately for three age groups: younger adults (<30 years old), middle-aged adults (30–64 years old), and older adults (≥65 years old) because each age group had different challenges at different points in the pandemic. Due to the large number of tests and difficulty replicating interaction effects, the p-value was set to < .01 for the moderation analysis.

### Results

Descriptive statistics for study variables are in Table 1. Table 2 reports results of the multilevel models that show personality change during the pandemic, accounting for the effect of time over the three assessments. Consistent with the previous studies on change in neuroticism early in the pandemic [14, 16], neuroticism was lower (approximately one-tenth of a standard

**Table 1. Descriptive statistics (mean [standard deviation] or % [sample size]) for study variables analyzed in the current study.**

| Variable | Mean (SD) or % (n) |
|---|---|
| Baseline age in years | 47.09 (15.77) |
| Age range | 18–109 |
| Gender (men) | 41.2% (2928) |
| Race (Black) | 8.3% (588) |
| Race (Asian) | 5.0% (354) |
| Race (Otherwise identified) | 9.1% (651) |
| Race (white) | 77.6% (5516) |
| Hispanic/Latino ethnicity (yes) | 17.5% (1241) |
| Education | 11.23 (2.27) |
| Time (years) | 3.48 (1.31) |
| UAS 1 Personality (May 2014-March 2018) | |
| Neuroticism | 21.22 (6.55) |
| Extraversion | 26.46 (6.35) |
| Openness | 36.32 (6.27) |
| Agreeableness | 36.38 (5.38) |
| Conscientiousness | 36.65 (5.43) |
| UAS 121 Personality (January 2018-April 2020) | |
| Neuroticism | 21.92 (6.52) |
| Extraversion | 25.91 (6.28) |
| Openness | 35.99 (6.23) |
| Agreeableness | 35.72 (5.47) |
| Conscientiousness | 35.91 (5.56) |
| UAS 237 Personality (April 2020-February 2022) | |
| Neuroticism | 21.73 (6.53) |
| Extraversion | 25.29 (6.25) |
| Openness | 35.34 (6.32) |
| Agreeableness | 35.27 (5.61) |
| Conscientiousness | 35.30 (5.70) |
| Pre-COVID (May 2014-Feburary 29, 2020) | |
| Neuroticism | 21.65 (6.54) |
| Extraversion | 26.13 (6.31) |
| Openness | 36.12 (6.25) |
| Agreeableness | 35.98 (5.44) |
| Conscientiousness | 36.20 (5.52) |
| COVID 2020 (March 1-December 31, 2020) | |
| Neuroticism | 21.52 (6.47) |
| Extraversion | 25.36 (6.28) |
| Openness | 35.25 (6.33) |
| Agreeableness | 35.37 (5.62) |
| Conscientiousness | 35.50 (5.63) |
| COVID 2021+ (January 1, 2021-February 16, 2022) | |
| Neuroticism | 22.31 (6.66) |
| Extraversion | 25.07 (6.15) |
| Openness | 35.59 (6.28) |
| Agreeableness | 35.01 (5.55) |

(*Continued*)

**Table 1.** (Continued)

| Variable | Mean (SD) or % (n) |
| --- | --- |
| Conscientiousness | 34.74 (5.83) |

*Note.* N = 7,109. UAS 1 *n* = 4,495. UAS 121 *n* = 7,019.

UAS 237 *n* = 7,109. Pre-Covid *n* personality assessments = 11,514. COVID 2020 *n* = 5224. COVID 2021+ *n* = 1885.
SD = standard deviation.

deviation) in 2020 compared to pre-pandemic levels. This decline, however, was not apparent in the next phase of the pandemic; neuroticism measured in 2021–2022 was not statistically different than neuroticism measured prior to the pandemic. Note that the time trend for neuroticism was positive, which indicated that neuroticism increased over time. The negative coefficient for COVID indicated a decrease in neuroticism during the pandemic, despite the time trend of increases over time. A different pattern emerged for the other four traits. For these traits, there was no difference in 2020 compared to their pre-pandemic levels. Extraversion, openness, agreeableness, and conscientiousness, however, declined in 2021–2022 compared to their level pre-pandemic.

A different pattern of personality change was apparent when the sample was split into three age groups (Table 3). The divergence by age was largest for neuroticism (Fig 1). When was measured in 2020, older adults had the greatest decline in neuroticism. Middle-aged adults also declined in neuroticism, with an effect size about half that of older adults. Younger adults showed this initial decline, but it was not statistically significant. The bigger discrepancy across age groups occurred for personality measured in 2021–2022. In this case, middle-aged adults continued to decline in neuroticism at this later stage of the pandemic, as did older adults, albeit the decline was not statistically significant. In contrast, younger adults had a significant increase in neuroticism in 2021–2022 compared to prior to the pandemic. The pattern that emerged for the remaining traits was similar across the four traits, with declines for both younger and middle-aged adults in 2021–2022. There were two patterns particularly worth noting. First, the coefficients for agreeableness and conscientiousness were at least twice as large among younger than middle-aged adults, which indicated larger declines in this age group. Second, there was no significant change in these traits among older adults in either 2020 or

**Table 2. Multilevel modeling of personality change from before to during the pandemic.**

| Model estimates | Neuroticism | Extraversion | Openness | Agreeableness | Conscientiousness |
| --- | --- | --- | --- | --- | --- |
| Fixed regression coefficients | | | | | |
| Intercept (SE) | 30.62 (.42)** | 24.71 (.43)** | 29.23 (.41)** | 33.41 (.35)** | 29.95 (.36)** |
| Linear slope (SE) | .19 (.03)** | -.23 (.02)** | -.22 (.03)** | -.23 (.03)** | -.22 (.02)** |
| COVID 2020 (SE) | -.62 (.10)** | -.10 (.08) | -.13 (.10) | .08 (.09) | -.09 (.09) |
| COVID 2021+ (SE) | -.17 (.12) | -.44 (.10)** | -.53 (.11)** | -.36 (.11)** | -.62 (.11)** |
| Random variances | | | | | |
| Variance intercept (SE) | 28.02 (.55)** | 31.26 (.58)** | 27.37 (.53)** | 18.93 (.39)** | 20.46 (.41)** |
| Variance linear slope (SE) | .16 (.02)** | .14 (.02)** | .13 (.02)** | .20 (.02)** | .16 (.02)** |
| Residual variance (SE) | 10.11 (.17)** | 7.26 (.12)** | 9.54 (.16)** | 9.04 (.15)** | 8.37 (.14)** |

*Note.* N = 7,109. COVID 2020 is personality assessments between March 1-December 31, 2020 and COVID 2021+ is personality assessments between January 1, 2021-February 16, 2022 both compared to pre-Covid (personality assessed prior to March 1, 2020. Analyses controlled for age, gender, race, Hispanic/Latino ethnicity, and education.

**p < .01.

Table 3. Effect of the pandemic on personality change by age group.

| Model estimates | Neuroticism | Extraversion | Openness | Agreeableness | Conscientiousness |
|---|---|---|---|---|---|
| Younger adults (<30) | | | | | |
| COVID 2020 (SE) | -.36 (.29) | .25 (.25) | .02 (.29) | .15 (.29) | -.11 (.29) |
| COVID 2021+ (SE) | .75 (.29)** | -.44 (.25) | -.55 (.28) | -.94 (.28)** | -1.23 (.28)** |
| Middle-aged adults (30–64) | | | | | |
| COVID 2020 (SE) | -.54 (.12)** | -.18 (.10) | -.24 (.11)* | .01 (.11) | -.13 (.10) |
| COVID 2021+ (SE) | -.40 (.14)** | -.44 (.12)** | -.59 (.14)** | -.30 (.14)* | -.50 (.13)** |
| Older adults (65+) | | | | | |
| COVID 2020 (SE) | -1.14 (.24)** | -.02 (.20) | .13 (.22) | .27 (.21) | .06 (.21) |
| COVID 2021+ (SE) | -.53 (.29) | -.22 (.24) | -.07 (.28) | .27 (.26) | -.08 (.25) |

*Note*. N = 7,109. Younger adult *n* = 1,105. Middle-aged adult *n* = 4,245. Older adult *n* = 1,759. Estimates control for normative age-related trends (i.e., slope component) and the sociodemographic covariates.

* $p < .05$.

** $p < .01$.

2021–2022: Extraversion, openness, agreeableness, and conscientiousness during the pandemic for participants over 65 were similar to pre-pandemic levels. The continuous interactions with age supported the overall pattern of age differences in personality change during the pandemic (S1 Table). Specifically, there was a negative interaction between COVID year and age on neuroticism for both 2020 and 2021–2022, which indicated the decline in neuroticism was larger at older ages in 2020 and the increase was larger at younger ages in 2021–2022, respectively. Likewise, the age interactions for agreeableness and conscientiousness indicated the decline in these two traits in 2021–2022 was stronger among relatively younger than relatively older participants. The interaction with age for 2020 for agreeableness was also significant.

Personality change during COVID was also moderated by Hispanic/Latino ethnicity (S1 Table). Hispanic/Latino participants did not experience the decline in neuroticism apparent among non-Hispanic/Latino participants. Hispanic/Latino participants also decreased more in agreeableness earlier in the pandemic than non-Hispanic/Latino participants. Both Hispanic/Latino and non-Hispanic/Latino participants declined in extraversion, openness, and conscientiousness in 2021–2022, but this decline was larger for Hispanic/Latino participants. There was less evidence for differences by the other sociodemographic groups (S1 Table).

## Discussion

Replicating previous work on personality change in the acute phase [14, 16], the present research found a significant decrease in neuroticism in 2020 compared to neuroticism prior to the pandemic. There was no significant change in the other traits in 2020. There was, however, a different pattern of change when personality was measured in 2021–2022: The beneficial effect of the pandemic on neuroticism dissipated, whereas there was significant decline in the other four traits compared to before the pandemic. Importantly, significant age differences also emerged that indicated that the decline in neuroticism in 2020 was largest for older adults, whereas the decline in the other four traits in 2021 was apparent in middle-aged and particularly younger adults. The present research thus suggests differential acute and longer-term time of measurement effects of the pandemic on personality change.

At the sample level, change in personality from before to during the pandemic was approximately one-tenth of a standard deviation. Although modest in absolute terms, it can be put in

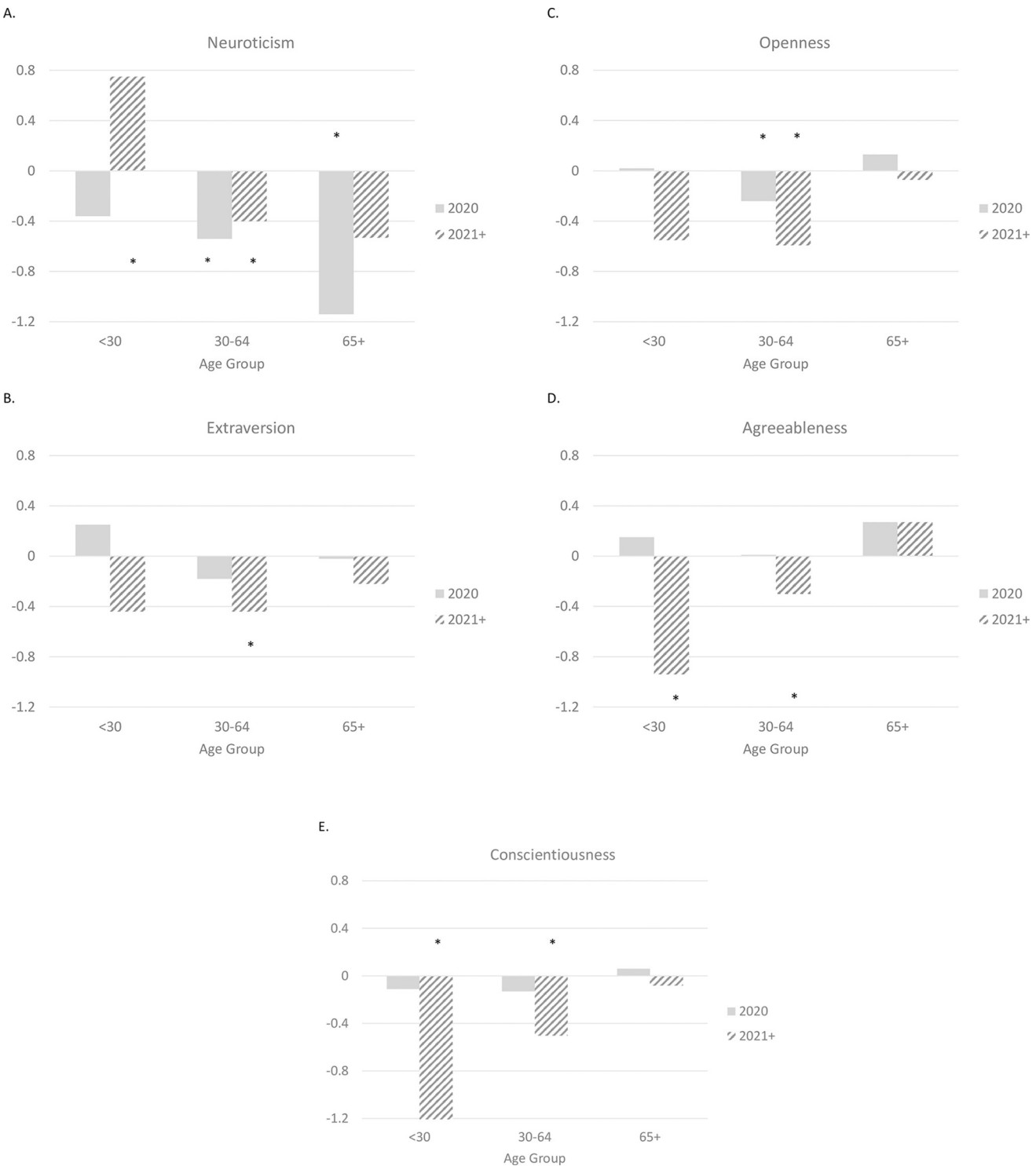

**Fig 1.** Age differences in the effect of the pandemic on personality change in 2020 and in 2021–2022 for neuroticism (Panel A), extraversion (Panel B), openness (Panel C), agreeableness (Panel D), and conscientiousness (Panel E). Asterisks indicate significant personality changes from pre-pandemic levels.

the perspective of developmental changes that occur over adulthood. Normative personality change has been estimated to be approximately one-tenth a standard deviation per decade in adulthood [17]. Given our analyses accounted for these normative age-related changes, the change observed during the short time of the pandemic approximated the degree of change usually observed over a decade. In addition, the changes were much larger for some demographic groups, including the decline in neuroticism for older adults, the decline in conscientiousness for younger adults, and the decline in extraversion for Hispanic/Latino participants, which were about one-fifth of a standard deviation.

The present research adds to the replicated evidence that neuroticism declined early in the pandemic [14, 16]. This decline is particularly surprising against the backdrop of other longitudinal research on mental health that found symptoms of depression, anxiety, and psychological distress increased during the first year of the pandemic [1, 2, 23]. These findings appear contradictory, particularly because symptoms of depression and anxiety are expressions of neuroticism [24]. Both changes, however, may occur simultaneously. It may be that, prior to the pandemic, individuals higher in neuroticism ascribed feelings of distress to this dispositional aspect of themselves. The fear and uncertainty caused by the pandemic, however, may have provided a reason for such feelings, leading to declines in perceptions of dispositional neuroticism. Further, prior to the pandemic, there were no behavioral suggestions to express or cope with neuroticism, but pandemic guidance (washing hands, social distancing, masking) gave people a preventive behavior to engage in against the external stressor. The messaging around taking care of one's mental health may also have contributed to decreases in neuroticism, especially for older adults since so much of the messaging was around taking care of this age group. It is also possible that the greater social cohesion early in the pandemic brought a sense of belonging that lessened a general disposition toward distress and/or observing the distress in the world had individuals re-evaluate their own tendency towards fear and anxiety. Further, there may be social comparison processes that shape how individuals perceive themselves. That is, ratings of personality are based, in part, on comparisons to other people. Early in the pandemic, when there was a lot of reporting on fear and anxiety about the virus in the media and on social media, individuals may have viewed themselves as less fearful and anxious than those around them: Individuals may have viewed themselves as less neurotic because the social norms around neuroticism shifted. Three studies now document this decline in neuroticism early in the pandemic.

A completely different pattern of change emerged during the adaptation phase of the pandemic. Neuroticism did not continue to decline, but rather was not statistically different from prior to the pandemic, which suggests the beneficial decline in neuroticism due to the pandemic was temporary. In addition, the other four traits, which did not change in the acute phase, all declined significantly in 2021–2022 compared to before the pandemic. This pattern suggests that for extraversion, openness, agreeableness, and conscientiousness, there was either a delayed effect that took longer to become apparent and/or different stressors and strains later in the pandemic contributed to these changes rather than the stressors and strains earlier in the pandemic. One possibility is that the social cohesion apparent early in the pandemic helped support stability of these traits. That is, in the acute phase, despite fear and uncertainty, the increase in social support [3] and sense of community [25] may have helped maintain personality. The decline in social support [26] and increase in social conflict on pandemic-related protective measures [27], may explain at least part of change observed in 2021–2022.

In our first paper on personality change very early in pandemic, we hypothesized a decrease in extraversion and conscientiousness because of restrictions on social gatherings and the loss of daily routines that often give structure to one's life, respectively. We did not, however, find any support for these declines [14]. The present analyses suggest a delayed or longer-term

effect on these traits. Early in the pandemic, there were anecdotal stories of long-lost connections being reestablished as old friends and acquaintances reached out to one another [28, 29]. Such connections may have helped support extraversion in the acute phase of the pandemic. Over a year of restrictions on social gatherings–either mandated or self-imposed over safety concerns–may have culminated in feeling less temperamentally outgoing than prior to the pandemic. Likewise, it might have taken more time for the lack of structure and fewer immediate responsibilities to consolidate into declines in conscientiousness. It may also be the case that, prior to the pandemic, external structures that supported schedules and routines were perceived as the individual's own level of conscientiousness. Without this stability and structure, it may be harder to be organized and follow through on responsibilities. The changes observed in 2021–2022 may be the accumulation of changes in daily life that took more time to culminate in trait decline.

There were also significant declines in openness and agreeableness. These declines may have been, in part, a response to the social upheaval in response to the pandemic that was sharper in 2021–2022. The continued uncertainty around the pandemic, particularly as it dragged into a second year [30], as well as the decline in mobility [31], may have led individuals to narrow their activities and worldviews. Likewise, there may have been a decrease in interest in art and artistic experiences because of less ability to engage in art due to closures of concert venues, museums, theaters, etc. The move to online communication and entertainment and reliance on social media may have decreased exposure to new ideas. Such changes may have contributed to declines in openness. There has been a decline in trust apparent for decades [17, 32]. Although there was an increase in confidence in science and the medical community early in the pandemic, this increase was short-lived and the decline precipitous during the second year of the pandemic [33]. The decline in agreeableness observed later in the pandemic is consistent with this trend. It is notable that this decline is apparent controlling for the general time trend of declines in agreeableness. This decline might have been partly fueled by amplification of mis/disinformation that undermines trust and may also highlight benefits to not being straightforward.

Two sociodemographic factors were significant moderators of personality change during the pandemic: age and Hispanic/Latino ethnicity. Compared to middle-aged and older adults, the personality of younger adults seemed particularly sensitive to change. Personality tends to develop most and consolidate during young adulthood [34], with the pattern of development toward greater maturity in the form of declines in neuroticism and increases in agreeableness and conscientiousness [35]. Over a year into the pandemic, however, young adults show the opposite of this developmental trend. The personality of older adults, in contrast, is thought to be more impervious to change (at least until very old age or cognitive impairment [36–38]); and, indeed, four of the five traits were relatively impervious to change among older adults. There may also be other reasons for the age differences in personality change. We cannot, for example, distinguish between age and cohort because they are confounded in the current sample; it is possible that the differences are due to cohort rather than age. It is also possible that different age groups faced different challenges in the second year of the pandemic, such as instability in the job market and school-related stressors (e.g., continued school closures, quarantining of the self or one's children after exposure). Such stressors may be more impactful for younger and middle-aged adults than older adults, who also may both be less likely to experience and have more resources to handle pandemic-related stressors that did occur.

Personality change during the pandemic was also moderated by Hispanic/Latino ethnicity. In 2020, Hispanic/Latino participants did not decrease in neuroticism but did decrease in agreeableness earlier than non-Hispanic/Latino participants. This pattern could be due, in part, to the strain of the pandemic not equally distributed across the population. The financial

cost of the pandemic was larger for Hispanic/Latino adults compared to their counterparts [39], and, at the same time, this population had higher rates of hospitalization and death due to COVID than non-Hispanic white adults [40]. Further, the decline in extraversion, openness, and conscientiousness in 2021–2022 was stronger for Hispanic/Latino participants than non-Hispanic/Latino participants. Perhaps these declines were because of processes that may have been apparent across the population were amplified by ongoing stressors of high-risk work situations and risk of COVID for themselves and their families. Surprisingly, although Black adults faced similar stressors, in this sample, Black participants did not show a similar pattern of personality change (i.e., the moderation analysis indicated no difference in change between Black and white participants).

The present research focused on personality change during the coronavirus pandemic. It is important to note other significant collective events in the United States during this time. The death of George Floyd, the subsequent social justice protests, the backlash to the protests, and the January 6, 2021 insurrection at the U.S. Capitol are significant events that occurred during this time that may also have shaped the observed changes. More research needs to tease apart whether/how different events may have shaped personality change.

## Implications for models of personality

There are several theoretical accounts to explain personality development across the lifespan. Biologically-oriented models indicate personality in adulthood is relatively impervious to environmental pressures and changes that are not biologically-based should rebound [34]. Environmental models, in contrast, highlight life events in the trajectory of adult personality, although evidence on specific life events tends to be mixed and sometimes conflicting [8]. The neuroticism finding represents a significant time of measurement effect that replicated across three studies. We are not aware of similar population-wide effects that have replicated across independent studies. The findings suggest that a large scale, global event had an impact on personality at the population level. It appears that this decline was transitory; it is too early to determine whether the changes observed in 2021–2022 will endure or dissipate with time. It is also possible personal experiences and perceptions of collective events may be more impactful on personality than the event itself [41].

Personality traits go through most development in adolescence and early adulthood and tend to reach stability about age 30 [34]. At the other end of adulthood, personality tends to remain stable until cognitive impairment reduces stability [36]. It is notable, but perhaps not surprising, that most significant personality change during the pandemic occurred in younger adulthood, with most traits showing no change among older adults. It is further of note that middle-aged adults were more similar to younger adults than older adults (except for neuroticism). It is unclear whether this pattern is due to greater malleability of traits earlier than later in adulthood or whether the stressors and strains of the pandemic, which differed across age groups, led to more personality change.

These findings may have implications for long-term outcomes associated with personality. Individuals higher in conscientiousness, for example, tend to achieve more education [42] and income [43], develop fewer chronic diseases [44], are at lower risk of dementia [45], and ultimately live longer [46]. The decline in conscientiousness, particularly for younger adults, may have consequences for these outcomes, especially if the decline is not transitory. Higher neuroticism is associated with engagement in health-risk behaviors [47, 48] and is a risk factor for poor mental health outcomes [24]. This increase may make some individuals more vulnerable to poor outcomes. It is especially worrying that the largest changes in these two traits were among younger adults, as the implications of these changes may ripple throughout their adult lives.

Although there should be some confidence in the decline in neuroticism, given that it has replicated, the other findings need to be interpreted with caution until replicated. It is of note that the lack of personality change in 2020 replicated our previous study on personality change in the acute phase of the pandemic (but Krautter and colleagues [16] found a decrease in extraversion during this time in a sample from Germany). Most importantly, the changes that occurred in 2021–2022 need to be replicated and put in the context of the sample. The sample was large and used a well-established measure of personality. The sample, however, was from the United States. No part of the world escaped the pandemic, but the course and response to the virus varied considerably across countries, and even within the same country. More research is needed to evaluate personality change during the pandemic in other cultural contexts and populations. In addition, although our sample was fairly diverse, the percentage of people of color was relatively low. The sample may have been underpowered to detect different patterns of personality change for people of color (the sample of Hispanic/Latino participants was larger and thus more powered to detect differences). This research documents personality traits over the first two years of the pandemic but the changes cannot be attributed solely to the pandemic. As discussed above, political and social upheaval co-experienced with the pandemic may have also contributed to the observed changes. We identified a time of measurement effect on change but were unable to distinguish the specific reasons for the changes. Pandemic-related policies and restrictions may be an additional contextual factor that is important for personality change. In the present research, participants were from around the United States who experienced very different state government responses to the pandemic (e.g., California versus Florida). Future research could address whether specific policy differences across states or countries have different impacts on change. Also, with few assessments of personality per participant, it was not possible to test for nonlinear changes over time. Future research would benefit from more assessments of personality to be able to test for such change. Further, there may be personality change related to infection with SARS-CoV-2, particularly for individuals with severe cases and/or long COVID. Recent evidence indicates significant changes in brain structure and cognitive function associated with SARS-CoV-2 infection (Douaud et al., 2022) [49]. Personality change could be one outcome of such alterations in neurological structure. The present research could not address this possibility. Finally, it would be worthwhile to take a "nuance" approach [50] and analyze change in specific items of the BFI to determine whether changes were driven by specific components of each trait.

Despite these limitations, the present research offers new evidence for longitudinal change in personality across the pandemic. This research highlights the need to continue to assess longitudinal changes, as the pandemic may have cumulative effects that were not apparent in the first few months. This research also highlights the differential impact on personality change across demographic groups (e.g., young adults, Hispanic/Latino). Future research needs to continue to track trends in personality change to evaluate potential longer-term outcomes associated with this change, particularly for groups impacted the most.

## Supporting information

**S1 Table. Interaction terms between sociodemographic factors and the pandemic on personality change.**
(DOCX)

**S2 Table. Syntax for reported analyses.**
(DOCX)

## Acknowledgments

The project described in this paper relies on data from survey(s) administered by the Understanding America Study (UAS), which is maintained by the Center for Economic and Social Research (CESR) at the University of Southern California (USC). The content of this paper is solely the responsibility of the authors and does not necessarily represent the official views of USC or UAS.

## Author Contributions

**Conceptualization:** Angelina R. Sutin, Yannick Stephan, Antonio Terracciano.

**Data curation:** Angelina R. Sutin, Antonio Terracciano.

**Formal analysis:** Angelina R. Sutin, Antonio Terracciano.

**Funding acquisition:** Angelina R. Sutin.

**Methodology:** Martina Luchetti, Damaris Aschwanden, Ji Hyun Lee, Amanda A. Sesker, Antonio Terracciano.

**Writing – original draft:** Angelina R. Sutin.

**Writing – review & editing:** Yannick Stephan, Martina Luchetti, Damaris Aschwanden, Ji Hyun Lee, Amanda A. Sesker, Antonio Terracciano.

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
