## [Decision Letter · Decision Letter 0]

25 Jul 2022

PONE-D-22-11403Differential personality change earlier and later in the coronavirus pandemic in a longitudinal sample of adults in the United StatesPLOS ONE

Dear Dr. Sutin,

Thank you for submitting your manuscript to PLOS ONE. After careful consideration, we feel that it has merit but does not fully meet PLOS ONE’s publication criteria as it currently stands. Therefore, we invite you to submit a revised version of the manuscript that addresses the points raised during the review process.

I recommend that it should be revised taking into account the changes requested by the reviewers. Since the requested changes include valuable comments, I would like to give you a chance to improve your manuscript.

We look forward to receiving your revised manuscript.

Kind regards,

Baogui Xin, Ph.D.

Academic Editor

PLOS ONE

Journal Requirements:

Reviewers' comments:

Reviewer's Responses to Questions

**Comments to the Author**

1. Is the manuscript technically sound, and do the data support the conclusions?

Reviewer #1: Yes

Reviewer #2: Yes

2. Has the statistical analysis been performed appropriately and rigorously? 

Reviewer #1: Yes

Reviewer #2: Yes

3. Have the authors made all data underlying the findings in their manuscript fully available?

Reviewer #1: Yes

Reviewer #2: Yes

4. Is the manuscript presented in an intelligible fashion and written in standard English?

Reviewer #1: No

Reviewer #2: Yes

5. Review Comments to the Author

Reviewer #1: Major comments:

Overall, I find this article to be an important addition to the literature due to its ability to validate two previous studies with a larger and more representative sample of Americans, and its ability to complement the growing literature around other aspects of social and mental health impacted by the COVID-19 pandemic. I find the methods to be strong, and clearly written, but could benefit from including additions as listed below. I found the discussion to be well written, and like the use of describing the phases of the pandemic as acute adjustment period, adaption phase, etc. I would encourage the authors to extend that terminology throughout the manuscript. Upon reading the entire manuscript, it seems as though there may have been multiple authors in the writing: the methods, results, and discussion sections were clear and well-written, but the introduction needs significant work to ensure clarity. A good edit by a single voice would benefit the manuscript.

Minor comments:

Abstract: If possible, please list the 5 personality traits assessed within the first 1-2 sentences. Since this journal hosts a broad range of topics, not all readers will be as familiar with these personality traits as the authors.

Introduction: Please provide additional context on the governmental policies and recommendations regarding COVID-19 that you feel may have impacted the personality traits you have assessed. There was wide variation across the world in these policies and recommendations, and brief description would benefit the reader.

Methods: Please include the anticipated ranges for the traits assessed within the BFI tool, and the directionality of scores (is higher indicating more or less neurotic trait? Etc.)

Table 1: Please include a more descriptive title for your table – tables should be able to stand on their own if ever used outside of the manuscript text.

Table 1: I believe “of” should be “or” in your third column heading (“mean (SD) OR % (n)”)

Table 1: consider plotting population averages of each personality trait on a figure to allow readers to quickly visualize potential change in population means across the time periods. It might also be helpful to include the years each wave was performed within the table or figure text (e.g. UAS1 (May 2014 – March 2018)).

Reviewer #2: Thank you for the opportunity to review this Interesting article entitled “Differential personality change earlier and later in the coronavirus pandemic in a longitudinal sample of adults in the United States“.

In this article, longitudinal data from the Understanding America Study (UAS) were used to examine personality changes at the beginning and later during the Covid-19 pandemic. Results indicate that there was a decrease in neuroticism at the beginning of the pandemic, but this was not present later in the pandemic. At this later time, however, there were decreases in the other four personality traits. In addition to these general trends, the authors also show that age plays a large role and that interactions with Hispanic/Latino ethnicity are present as well.

The article is written very well and in a comprehensible way. The authors make clear what they have studied, how they can explain their findings, and what the implications are. Overall, I think the article is already very good in its current form, however, I would like to make a few suggestions for changes. I hope that my comments can be helpful to the authors to further improve their work.

Introduction:

- The authors briefly state that there are few studies (2 are mentioned) on other collective stressful events and subsequent personality changes. I suggest specifying the results of these studies and relating them to the studies that were carried out during the Covid-19 pandemic.

- In addition, it is mentioned that personality aspects change by one tenth of a standard deviation in a decade. In my opinion, more could be said in the introduction about which changes are normal over certain periods of time and whether these changes concern all aspects of personality equally. As age becomes very important in the course of the article, I would also elaborate on how much personality changes in early, middle, and late adulthood. If there is literature on this, it could also be addressed which aspects of personality are particularly susceptible to change at which stages of life. This could then also be taken up in the discussion and compared with the results.

Materials and Methods

- I am not sure if I have understood this correctly. But I was surprised that with such a large sample there was only one measurement per person during Covid (either at the beginning or from 2021). I would recommend highlighting more clearly why it was not possible to have three measurements per individual (pre-pandemic, early pandemic and from 2021 onwards). The fact that they are not the same people at the beginning and later in the pandemic is mentioned once, but I would emphasize this more. In addition, I would recommend making comparisons between these two samples to show that they were comparable based on, for example, age, demographics, and baseline scores on personality aspects.

- Personality scale scores were calculated as sums. Since this could be problematic in case of missing values, I would recommend inserting a short statement about missing values or to use average values.

- I like that the authors give test-retest correlations. Since the article is about changes in these scales, it would be interesting to know whether these correlations differ in comparison to the validation study of the scale or other longitudinal studies on these peronality aspects.

- In my opinion, the choice of covariates should be briefly justified.

- It was not clear to me when reading how the interactions (with the exception of the different models for the three age groups) were calculated. Were they all included together in one model, or were there separate models per interaction?

Results:

- In Table 1, I recommend including not only the age range, but also the mean and standard deviation. In addition, I would indicate the range of years. Also, in this table I wondered why the personality scores were given for the three UAS waves and not for the time points relevant for the study (before the pandemic, from March 2020, from 2021).

- Regarding Table 2, I wondered why a linear effect of time was assumed. I recommend specifying whether, for example, quadratic or cubic trends were also tested. It would also be interesting to see a graph showing the fluctuations of each personality aspect over time, perhaps this would show some key moments where changes occurred and provide some possible further explanations of the results. In addition, I found it confusing that change over time was positive for Neuroticism, while changes toward Covid 2020 and Covid 2021+ were negative. Perhaps this was a typo or there is a simple explanation.

- Figure 1: To highlight which of the changes were significant, I suggest using asterisks.

Discussion:

- As very different effects were shown for the three age groups, I suggest briefly mentioning age effects in the first paragraph where the results are summarized.

- The decrease in neuroticism at the beginning of the pandemic could be discussed in more detail on page 17, perhaps with more than one possible explanation. In particular, the finding that this was especially true for older adults could be explained more precisely.

- In general, I would appreciate it if age differences in relation to the individual personality aspects could be elaborated in more detail.

- To understand the changes in the individual personality aspects, it might be worthwhile to take a closer look at the items of the scale used. It is possible that the changes are driven primarily by items that ask about behaviours that were not possible or allowed at certain times during the pandemic. For example, the item "Is a reliable worker" might not be answered well if someone can no longer work because of a curfew. Also, the items "is talkative"/” Is outgoing, sociable” could potentially be problematic, if asked during a national curfew.

6. PLOS authors have the option to publish the peer review history of their article (what does this mean?). If published, this will include your full peer review and any attached files.

Reviewer #1: No

Reviewer #2: No

---

## [Author Response · Author response to Decision Letter 0]

18 Aug 2022

Response to Reviewers

Reviewer #1

Overall, I find this article to be an important addition to the literature due to its ability to validate two previous studies with a larger and more representative sample of Americans, and its ability to complement the growing literature around other aspects of social and mental health impacted by the COVID-19 pandemic. I find the methods to be strong, and clearly written, but could benefit from including additions as listed below. 

We thank the Reviewer for the overall positive evaluation of our manuscript and the suggestions for improvement.

I found the discussion to be well written, and like the use of describing the phases of the pandemic as acute adjustment period, adaption phase, etc. I would encourage the authors to extend that terminology throughout the manuscript. Upon reading the entire manuscript, it seems as though there may have been multiple authors in the writing: the methods, results, and discussion sections were clear and well-written, but the introduction needs significant work to ensure clarity. A good edit by a single voice would benefit the manuscript.

We now use the suggested terminology throughout the manuscript. We also rewrote part of the Introduction to ensure clarity and a single voice throughout the manuscript.

Minor comments:

Abstract: If possible, please list the 5 personality traits assessed within the first 1-2 sentences. Since this journal hosts a broad range of topics, not all readers will be as familiar with these personality traits as the authors.

We added the five traits to the first sentence of the Abstract.

Introduction: Please provide additional context on the governmental policies and recommendations regarding COVID-19 that you feel may have impacted the personality traits you have assessed. There was wide variation across the world in these policies and recommendations, and brief description would benefit the reader.

We now mention some restrictions that may have had an impact on extraversion in the Introduction (p. 5). We refrain, however, from an in-depth consideration of governmental policies and recommendations to reduce the spread of COVID-19 because participants in this sample were from across the United States, and states had very different responses to COVID-19 (e.g., mask mandates and venue closures in California and New York versus mandates against masks and closures in Florida and Texas). Because we are not able to rigorously test whether such policies shaped personality change, we do not address the issue in depth in the Introduction. We do now discuss government policies and recommendations as an additional contextual factor that may contribute to personality change (p. 27).

Methods: Please include the anticipated ranges for the traits assessed within the BFI tool, and the directionality of scores (is higher indicating more or less neurotic trait? Etc.)

We added the possible ranges for each trait and the directionality of the scores to the Method (p. 9). 

Table 1: Please include a more descriptive title for your table – tables should be able to stand on their own if ever used outside of the manuscript text.

We revised the title of Table 1 to be more descriptive.

Table 1: I believe “of” should be “or” in your third column heading (“mean (SD) OR % (n)”)

Yes, thank you for catching the typo. It has been corrected.

Table 1: consider plotting population averages of each personality trait on a figure to allow readers to quickly visualize potential change in population means across the time periods. It might also be helpful to include the years each wave was performed within the table or figure text (e.g. UAS1 (May 2014 – March 2018)).

We reformatted Figure 1 to highlight the changes in personality across time periods. We added the years for each wave to Table 1.

Reviewer #2

Thank you for the opportunity to review this Interesting article entitled “Differential personality change earlier and later in the coronavirus pandemic in a longitudinal sample of adults in the United States“. In this article, longitudinal data from the Understanding America Study (UAS) were used to examine personality changes at the beginning and later during the Covid-19 pandemic. Results indicate that there was a decrease in neuroticism at the beginning of the pandemic, but this was not present later in the pandemic. At this later time, however, there were decreases in the other four personality traits. In addition to these general trends, the authors also show that age plays a large role and that interactions with Hispanic/Latino ethnicity are present as well. The article is written very well and in a comprehensible way. The authors make clear what they have studied, how they can explain their findings, and what the implications are. Overall, I think the article is already very good in its current form, however, I would like to make a few suggestions for changes. I hope that my comments can be helpful to the authors to further improve their work.

We thank the Reviewer for the overall positive evaluation of our manuscript and the suggestions for improvement.

Introduction:

- The authors briefly state that there are few studies (2 are mentioned) on other collective stressful events and subsequent personality changes. I suggest specifying the results of these studies and relating them to the studies that were carried out during the Covid-19 pandemic.

We now describe the results of these two studies in the Introduction (pp. 4-5).

- In addition, it is mentioned that personality aspects change by one tenth of a standard deviation in a decade. In my opinion, more could be said in the introduction about which changes are normal over certain periods of time and whether these changes concern all aspects of personality equally. As age becomes very important in the course of the article, I would also elaborate on how much personality changes in early, middle, and late adulthood. If there is literature on this, it could also be addressed which aspects of personality are particularly susceptible to change at which stages of life. This could then also be taken up in the discussion and compared with the results.

We added a paragraph on normative personality development and differences across age groups to the Introduction (pp. 6-7).

Materials and Methods

- I am not sure if I have understood this correctly. But I was surprised that with such a large sample there was only one measurement per person during Covid (either at the beginning or from 2021). I would recommend highlighting more clearly why it was not possible to have three measurements per individual (pre-pandemic, early pandemic and from 2021 onwards). The fact that they are not the same people at the beginning and later in the pandemic is mentioned once, but I would emphasize this more. In addition, I would recommend making comparisons between these two samples to show that they were comparable based on, for example, age, demographics, and baseline scores on personality aspects.

The number of assessments of personality available is due to the sampling structure of UAS; we used all data available on personality in the analysis. We now state this information in the description of the sample (p. 8). We also report the sociodemographic differences and differences in baseline personality between participants who reported on their personality in 2020 versus 2021-22 (p. 10).

- Personality scale scores were calculated as sums. Since this could be problematic in case of missing values, I would recommend inserting a short statement about missing values or to use average values.

We agree with the Reviewer that sums can sometimes be problematic for missing data. In the UAS, however, in each of the three assessments of personality, more than 99% of participants who completed the survey had personality scores, which indicates that missing data is not a significant concern. We note this information in the Method (p. 9).

- I like that the authors give test-retest correlations. Since the article is about changes in these scales, it would be interesting to know whether these correlations differ in comparison to the validation study of the scale or other longitudinal studies on these personality aspects.

The re-test correlations are similar to what has been reported in the past for the Big Five Inventory (p. 10).

- In my opinion, the choice of covariates should be briefly justified.

We briefly justify the covariates (p. 10).

- It was not clear to me when reading how the interactions (with the exception of the different models for the three age groups) were calculated. Were they all included together in one model, or were there separate models per interaction?

We clarify that the interactions were tested in separate models (p. 12).

Results:

- In Table 1, I recommend including not only the age range, but also the mean and standard deviation. In addition, I would indicate the range of years. Also, in this table I wondered why the personality scores were given for the three UAS waves and not for the time points relevant for the study (before the pandemic, from March 2020, from 2021).

We include the mean and standard deviation of age and added the range of years to Table 1. We also added the mean of personality from before the pandemic, from 2020, and from 2021-22 to Table 1.

- Regarding Table 2, I wondered why a linear effect of time was assumed. I recommend specifying whether, for example, quadratic or cubic trends were also tested. It would also be interesting to see a graph showing the fluctuations of each personality aspect over time, perhaps this would show some key moments where changes occurred and provide some possible further explanations of the results. In addition, I found it confusing that change over time was positive for Neuroticism, while changes toward Covid 2020 and Covid 2021+ were negative. Perhaps this was a typo or there is a simple explanation.

It was not possible to test for quadratic or cubic trends because there were too few assessments of personality. We agree with the Reviewer that nonlinear change is important to test, and we now discuss this limitation (p. 27). We reformatted Figure 1 to highlight the changes in personality across time periods for each trait. We also now explain why there is a difference in sign for the time trend versus the COVID coefficients for neuroticism (pp. 12-13).

- Figure 1: To highlight which of the changes were significant, I suggest using asterisks.

We added asterisks to highlight the significant changes in Figure 1.

Discussion:

- As very different effects were shown for the three age groups, I suggest briefly mentioning age effects in the first paragraph where the results are summarized.

We added a brief summary of the age effects in the first paragraph of the Discussion (p. 19).

- The decrease in neuroticism at the beginning of the pandemic could be discussed in more detail on page 17, perhaps with more than one possible explanation. In particular, the finding that this was especially true for older adults could be explained more precisely.

We provide more possible explanations for the decrease in neuroticism at the beginning of the pandemic (pp. 20-21).

- In general, I would appreciate it if age differences in relation to the individual personality aspects could be elaborated in more detail.

We now discuss more possible reasons for the age differences found in the second year of the pandemic (p. 24).

- To understand the changes in the individual personality aspects, it might be worthwhile to take a closer look at the items of the scale used. It is possible that the changes are driven primarily by items that ask about behaviours that were not possible or allowed at certain times during the pandemic. For example, the item "Is a reliable worker" might not be answered well if someone can no longer work because of a curfew. Also, the items "is talkative"/” Is outgoing, sociable” could potentially be problematic, if asked during a national curfew.

We agree that it would be important to follow up the findings reported in this manuscript with a “nuance” approach to personality to identify how specific items changed during the pandemic. We now indicate this question should be addressed in future work on personality change during the pandemic (p. 28).

---

## [Editor Report · Decision Letter 1]

30 Aug 2022

Differential personality change earlier and later in the coronavirus pandemic in a longitudinal sample of adults in the United States

PONE-D-22-11403R1

Dear Dr. Sutin,

We’re pleased to inform you that your manuscript has been judged scientifically suitable for publication and will be formally accepted for publication once it meets all outstanding technical requirements.

Kind regards,

Baogui Xin, Ph.D.

Academic Editor

PLOS ONE
---

## [Editor Report · Acceptance letter]

1 Sep 2022

PONE-D-22-11403R1 

Differential personality change earlier and later in the coronavirus pandemic in a longitudinal sample of adults in the United States 

Dear Dr. Sutin:

I'm pleased to inform you that your manuscript has been deemed suitable for publication in PLOS ONE. Congratulations! Your manuscript is now with our production department. 

Kind regards, 

on behalf of

Professor Baogui Xin 

Academic Editor

PLOS ONE